# Adsorption of Deoxynivalenol (DON) from Corn Steep Liquor (CSL) by the Microsphere Adsorbent SA/CMC Loaded with Calcium

**DOI:** 10.3390/toxins12040208

**Published:** 2020-03-25

**Authors:** Ahmed Shalapy, Shuangqing Zhao, Chenxi Zhang, Yifei Li, Hairong Geng, Sana Ullah, Gang Wang, Shujian Huang, Yang Liu

**Affiliations:** 1Key Laboratory of Agro-products Quality and Safety Control in Storage and Transport Process, Ministry of Agriculture and Rural Affairs/Institute of Food Science and Technology, Chinese Academy of Agricultural Sciences, Beijing 100193, China; 2018Y90100052@caas.cn (A.S.); 82101176187@caas.cn (S.Z.); Zhangchenxi@caas.cn (C.Z.); 82101186201@caas.cn (Y.L.); genghr2015@163.com (H.G.); 2017y90100154@caas.cn (S.U.); wanggang02@caas.cn (G.W.); 2School of Life Science and Engineering, Foshan University, Foshan, Guangdong 528231, China; sjhuang.foshan@163.com

**Keywords:** deoxynivalenol, adsorption, sodium alginate, carboxymethylcellulose, calcium chloride, corn steep liquor, sodium hydroxide

## Abstract

The occurrence of deoxynivalenol (DON) in animal feed is a serious issue for the livestock industry. Approaches using mycotoxin adsorbents are key to decreasing mycotoxin carryover from contaminated feed to animals. In this paper, a novel functional microsphere adsorbent comprising an alginate/carboxymethyl cellulose sodium composite loaded with calcium (SA/CMC-Ca) was prepared by an emulsification process to adsorb DON from polluted corn steep liquor (CSL) containing DON at a concentration of 3.60 μg/mL. Batch experiments were conducted under different experimental conditions: CSL volumes, reaction times, desorption times, and microsphere recyclability. Results showed that 5 g of microspheres reacted with 5 mL of DON-polluted CSL for 5 min, the microspheres can be recycled 155 times, and the maximum DON adsorption for the microspheres was 2.34 μg/mL. During recycling, microspheres were regenerated by deionized water every time; after the microspheres were cleaned, DON in the deionized water was degraded by sodium hydroxide (NaOH) at 70 °C for 1 h at pH 12. The mechanism for physical adsorption and hydrogen bonding was analyzed by scanning electron microscopy (SEM) and Fourier transform infrared spectrometry (FTIR). To the best of our knowledge, this is the first report showing that the microsphere adsorbent SA/CMC-Ca adsorbs DON. Therefore, we suggest that using microsphere absorbents would be a possible way to address DON-contaminated CSL issues in animal feed.

## 1. Introduction

Deoxynivalenol (DON), a type B trichothecene, is a secondary toxic compound generated by *Fusarium* species and has an epoxide group between C12 and C13 that represents various toxic effects of this molecule (Figure 1) [1]. On a global level, DON is believed to be one of the most hazardous and widespread polluting toxins to contaminate food and feed [2]. Cereals such as wheat, barley, maize, oats, and rye are mainly contaminated with DON [3]. DON can cause gastrointestinal dysfunction in monogastric animals (i.e., vomiting, diarrhea, and refusal to eat) and result in decreased animal performances [4,5]. High doses of DON immunosuppression in animals can cause circulatory shock and can even lead to death [6]. Thus, DON affects the production performance of livestock and, ultimately, results in economic losses.

Methods for the decontamination and reduction of DON in food and feed sources have been widely investigated [7]. Several methods controlling DON contamination encompass physical, chemical, and biological approaches [8,9,10]. Among them, mycotoxin adsorbents in feed are widely used for DON decontamination [11]. There are several types of adsorbents, such as bentonite, zeolite [12], activated charcoal [13], and yeast cell wall [14]. Although many DON absorbents are available, the low DON adsorption efficiencies, high costs, and pH limitations make their application in food and feed less efficient [15].

Sodium alginate (SA) is a natural gelling agent derived from the cell wall of *Sargassum* or *Turbinaria* brown algae [16]. SA is a polymeric flocculant with high adsorption capacities. It can interfere with many crosslinkers such as calcium(II) [17] and N,N’-methylenebisacrylamide (MBA) [18]. When SA crosslinks with calcium, a large amount of water is adsorbed and locked into the “eggbox” internal structure to generate a stable hydrogel; “eggbox model” (Figure 2). Carboxymethyl cellulose (CMC) is a derivative of cellulose and is formed by the reaction of cellulose with sodium hydroxide and chloroacetic acid [19]. As part of its macromolecular chain structure, CMC has many active groups, such as hydroxyl and carboxyl groups [20]. It is widely used in detergents, textiles, paper, foods, oil well drilling operations, and other industries as a thickening agent and flocculant [21,22]. Microspheres for SA/CMC-Ca can be prepared by two methods: extrusion and emulsification [23,24], and they are widely used for heavy metal removal from aqueous solutions [25]. However, there is a dearth of information on the use of SA/CMC for mycotoxin removal.

Corn steep liquor (CSL) is a by-product of corn starch preparations from wet-milling [26]. It contains high levels of amino acids, vitamins, and growth factors that play important roles in the feed industry [27]. Most of the DON in raw corn is transferred to CSL in the first stage of wet-milling. From a study in Korea, the mean contamination level of DON in CSL was 7.417 μg/g [28], In New Zealand, DON was found at high concentrations up to 8.8 μg/g in commercial wet-milling process of CSL [29]. The occurrence of DON in CSL is a severe issue for the livestock industry.

Based on previous experimental studies, SA/CMC-Ca microspheres were prepared by cross-linking and beading SA and CMC in CaCl_2_. The purpose of this study was to (1) synthesize a novel SA/CMC-Ca adsorbent composite microsphere with an “eggbox” structure to adsorb DON from CSL, (2) optimize microsphere adsorption and desorption parameters, (3) detoxify DON in microsphere cleaning water, (4) examine the recyclability of microspheres for adsorbing DON, and (5) analyze microsphere adsorption of DON in CSL by using scanning electron microscopy (SEM) and Fourier transform infrared spectrometry (FTIR).

## 2. Results

### 2.1. Microsphere DON Adsorption Test

It was evident from the observation that there were differences in DON peak areas between the control and treatment groups in Figure 3. The concentration of DON was calculated by the standard curve of DON. In the control group, the original DON concentration of CSL was 3.60 μg/mL, while in the treatment group, it was only 1.73 μg/mL. In general, after reacting with microspheres, nearly 50% of DON in CSL was adsorbed by the microspheres. The following results suggest that microsphere adsorbents can adsorb DON from CSL. 

### 2.2. The Effect of CSL Volumes

The adsorption ability of DON from different CSL volumes with 5 g of microspheres is shown in Figure 4. The results indicated that DON adsorption by the microspheres decreased as CSL volumes increased. In statistical analyses, CSL volumes of 3 mL, 4 mL, and 5 mL made no significant differences. However, when CSL volumes were increased up to 6 mL, 7 mL, 8 mL, 9 mL, and 10 mL, DON adsorption for microspheres decreased. Microspheres at 5 g can adsorb maximum DON from 5 mL CSL, and after that, the adsorption of DON by microspheres decreased. So, for 5 g microspheres, 5 mL CSL was the ideal volume to adsorb DON.

### 2.3. The Effect of Reaction Times 

The reaction time of microspheres in CSL also plays an important role in DON adsorption. The effects of microsphere reaction time with CSL on DON levels are shown in Figure 5. DON levels in CSL decreased as the reaction time of the microsphere adsorbent increased in CSL. According to the data, 48 h was the best reaction time of microspheres with CSL to decrease the DON level up to 1.26 μg/mL, while DON level was 1.82 μg/mL after 5 min of reaction time of microspheres with CSL. However, the speed of adsorption was different. For 48 h of reaction time, the average speed of adsorption was 0.00086 μg/mL per minute. For 5 min of reaction time, the speed of adsorption was 0.382 μg/mL per minute. There was a huge difference in adsorption speed for the two reaction times (48 h and 5 min), which meant that the microspheres adsorbed DON in CSL at the beginning quickly and the adsorption gradually slowed down as reaction time increased. Finally, adsorption reached a dynamic equilibrium. For 1 min and 3 min, reaction time was so short that the microspheres could not adsorb much DON. In the present study, considering the adequate reaction time of microspheres with DON-contaminated CSL, 5 min reaction time was selected for the next experimental work.

### 2.4. The Effect of Time for Desorption 

Deionized water was used to regenerate the microspheres after reacting them with DON-contaminated CSL, and the effect of desorption time on DON levels in the water was investigated. For microsphere desorption, the best desorption time was studied by testing DON levels in deionized water (Figure 6). The level of DON in water varied between 0.40 and 0.45 μg/mL for desorption times of 5 min or more, and the concentration of DON fluctuated at 0.1–0.2 μg/mL and 0.2–0.3 μg/mL, when the desorption times were 1 min and 3 min, respectively. When microspheres are kept for a long duration of time in deionized water, there are more chances that the microspheres will break. Therefore, 5 min was the best time to remove an adequate concentration of DON from microspheres.

### 2.5. Detoxification of DON in Microsphere Cleaning Water

To avoid the contamination of environment caused by DON in contaminated water, an alkaline and high-temperature environment was created for the degradation of DON in contaminated water. In Figure 7, the results showed DON detoxification rates up to 48.83%, 80.45%, and 98.67% for the different reaction times of 10 min, 30 min, and 60 min, respectively. Therefore, DON could be removed from contaminated water, which would not lead to environmental pollution after discharge. 

### 2.6. Microsphere Recyclability

The recyclability of the microspheres was evaluated by the DON level in CSL after reaction with the microspheres. Results are shown in Figure 8. For the first recycle, DON level in CSL was 2.62 μg/mL. For 5, 10, 15, 20, 25, 30, and 35 recycle times, DON levels in CSL after reaction with the microspheres were 2.35 μg/mL, 2.32 μg/mL, 2.24 μg/mL, 2.21 μg/mL, 2.05 μg/mL, 1.95 μg/mL, and 1.84 μg/mL, respectively. DON can physically adsorb onto the microspheres because of the high porosity of the microspheres. After recycling for 95 times with the microspheres, DON level in CSL was only 1.39 μg/mL. However, for 160, 165, and 170 recycling times, DON levels in CSL were 2.84 μg/mL, 3.24 μg/mL, and 3.37 μg/mL; and at that time, the microspheres had ruptured and were very soft. Consequently, the adsorption of CSL by microspheres can be performed and repeated for 155 times on a small scale with relatively good results. 

### 2.7. SEM Analysis 

Scanning electron microscopy (SEM) micrographs of microspheres are shown in Figure 9A–D. The SEM image in Figure 9A shows the outer surface of an SA/CMC-Ca microsphere. Figure 9B shows the internal interconnection of SA and CMC. The interwoven matrix created bright, large, irregular pore structures ideal for DON adsorption. In Figure 9C, after DON adsorption, the inner surface of the microspheres had smaller pores and darker color when compared with SA/CMC-Ca microspheres before DON adsorption. Figure 9D shows that cleaning the SA/CMC-Ca microspheres in deionized water restored the roughness to the internal surface of the SA/CMC-Ca microspheres similar to that observed in Figure 9B; and the color of the cleaned microspheres was also lighter than that observed in Figure 9C.

### 2.8. Adsorption Mechanism

FTIR can be used to analyze the functional groups of biopolymers and to predict molecular interactions. As shown in Figure 10, the presence of peaks in the region of 3334–3377 cm^−1^ showed the stretching vibration of O-H. The stretching vibration peak of O-H in SA-Ca microspheres was observed at 3377.24 cm^−1^ (Figure 10A). Compared to SA-Ca microspheres, the peak for stretching vibration of O-H in the SA/CMC-Ca microspheres became wider and moved to a lower wavelength at 3350.24 cm^−1^ (Figure 10B), which indicated that CMC might be hydrogen-bonded with SA inside the microspheres. For the SA-Ca microspheres, a stretching vibration peak of saturated C-H at 2854.55 cm^−1^ and 2922.06 cm^−1^ was observed. However, for the SA/CMC-Ca microspheres, the stretching vibration peak of C-H disappeared, indicating the participation of C-H in the cross-linking reaction.

The FTIR spectra of the SA/CMC-Ca microspheres before and after DON adsorption are shown in Figure 10B,C. The O-H and COO− wavelengths for microspheres after DON adsorption were longer to a certain extent than those before DON adsorption, which indicated that COO- in microspheres may be hydrogen-bonded with O-H in DON. For the SA/CMC-Ca microspheres that adsorbed DON, the stretching vibration peak of C-H at 2850.69 cm^−1^ and 2916.27 cm^−1^ appeared, indicating that the C-H bond for methyl (CH_3_-) and methylene (-CH_2_-) groups in DON appeared in microspheres. DON was adsorbed by the SA-CMC-Ca microspheres.

For the SA/CMC-Ca microspheres that desorbed DON, the wavelengths for O-H and COO− in the SA/CMC-Ca microspheres after DON desorption were shorter to a certain extent (Figure 10D) than those of microspheres that adsorbed DON (Figure 10C). This might be due to water adsorbed by the microspheres, and the peak intensity for the stretching of O-H increased significantly. Some of COO- in the SA/CMC-Ca microspheres, after DON desorption, no longer bonded to Ca^2+^_,_ which increased the COO- peak intensity.

## 3. Discussion

DON is highly cytotoxic and can suppress immune function, thus posing a threat to human and animal health. This is the first study to report the ability of SA/CMC-Ca microspheres to adsorb DON in CSL. Previously, SA/CMC-Ca microspheres have been used for the adsorption of lead. A static adsorption experiment demonstrated that lead ion adsorption of SA/CMC-Ca microspheres exceeded 99% under optimized conditions [30]. Besides, SA/CMC-Ca microspheres can also have a strong adsorption capacity for uranium and fluorine [25].

SA/CMC-Ca microspheres, however, could not degrade or bind other mycotoxins, such as zearalenone, aflatoxin B1, ochratoxin, fumonisin, and T-2 mycotoxins G1 and G2. Further experiments on adding different materials based on SA/CMC-Ca microspheres to adsorb various mycotoxins are being conducted in our laboratory.

The reaction system was at a ratio of 1:1 weight per volume (5 g microspheres in 5 mL CSL). We aimed to industrially apply microspheres to adsorb DON from CSL. The ideal reaction conditions were fewer microspheres and more CSL. However, from the data for the SA/CMC-Ca microspheres’ ability to adsorb DON, 5 g microspheres in 5 mL CSL was an appropriate condition for our experiments. Considering the cost of adsorbing DON in CSL for large-scale applications, we recycled the microspheres through adsorption-desorption-regeneration to reduce industrial costs. Similar strategies can also be found in chitosan-epichlorohydrin-triphosphate adsorbents for adsorption and desorption of Cu(II), Cd(II), and Pb(II) ions in an aqueous solution [31].

The reaction time was an effective factor during the reaction of microspheres with DON-polluted CSL. With the increase in reaction time, the adsorption of DON increased first and then decreased slowly, and it did not increase after almost 1 h of adsorption saturation. Similarly, the adsorption of Maxilon Blue GRL dye from an aqueous solution was studied at different contact times from 0 to 120 min. The results also showed that adsorption increased with increased times, and reached equilibrium at 60 min [32]. Sludge from an iron-ore processing area was used as an adsorbent to remove As, Mn, Zn, Cd, and Pb from aqueous solutions and it showed that adsorption capacities increased with contact times and equilibrium was reached in a single-metal experiment after 6 h and in mixed-metal experiments after 1 h [33].

Considering that microspheres break easily over longer times, the data showed that cleaning times of 1 min and 3 min were too short to adsorb enough DON, thereby limiting microsphere regeneration. Therefore, 5 min was the ideal desorption time. Similarly, regeneration research showed that the use of both distilled water and 30% H_2_O_2_ in 0.5 M HNO_3_ was highly effective in removing adsorbed arsenic from coconut husk carbon for up to three cycles [34]. Saturated activated fly ash and fresh activated fly ash can be regenerated with 150 mL of 1N NaOH for one day and activated fly ash showed higher desorption efficiencies of over 85% for the removal of chromium(VI) [35]. Deionized water and 0.1 M HCl were used for the desorption of heavy metals from petiolar felt-sheath of palm (PFP) for up to three cycles with desorption efficiencies of over 90% [36].

DON in cleaning water is also a severe problem for the ecosystem. We found that DON was degraded in water for 1 h in an alkaline environment at 70 °C. This alkaline degradation of DON was similar to that observed in a previous study with alkaline-degraded DON by heat treatment with 100% detoxification [37], which implied that DON had undergone structural changes in an alkaline environment and the reaction was accelerated by the high temperature. Similar work also found that DON decreased from 4.46 μg/g to 1.27 μg/g in the process of making tortillas from corn, which could be attributed to Ca(OH)_2_ treatment [38]. The safety of the degradation products of DON under alkaline conditions is also a concern. Five degradation products, namely, isoDON, NorDON A, NorDON B, NorDON C, and DON lactone, were isolated after harsh thermal or alkaline treatment of DON [37,39,40]. These degradation products were less cytotoxic compared to DON in an immortalized human kidney epithelial cell (IHKE) culture assay, in which DON revealed a median effective concentration (EC_50_) at approximately 1.1 mol/L, whereas NorDON A, B, and C did not exert any significant effect up to 100 mol/L [41]. In fact, we still think that alkaline and thermal degradation of DON is not the best method because of the difficulty of treating alkaline wastewater and the high cost of high temperature. Further, biodegradation of DON in water by the bacterium *Devosia* degradation is underway in our lab.

Recyclability of microspheres is an essential indicator for measuring industrial application. The microspheres were recycled for 170 times, among which 155 times showed a high adsorption capacity for DON. In the beginning, the microspheres did not adsorb much DON. This may have occurred as a result of the process of making microspheres because soybean oil on the surface of the microspheres interferes with their adsorption of DON. With recycle times increased, DON adsorption also increased. Microspheres were swollen after the increase in recycle times, the inner surface area of the microspheres was increased, and there were a lot of pore channels inside the microspheres. After 155 cycles, it was found that the microspheres became soft and began to break. This phenomenon was in contrast with another study, which showed that five consecutive adsorption/desorption cycles caused the adsorption capacity of uranium to decrease by 10% [42]. Glutaraldehyde-crosslinked metal-complexed chitosan can be reused to adsorb copper(II), zinc(II), nickel(II), or lead(II) ions up to three times, and the adsorbed amount of lead ion was 100 mg/g, 60 mg/g, and 30 mg/g as the reuse time increased [43]. For cross-linked xanthate, a chitosan adsorbent that decontaminates lead from aqueous media, adsorption-desorption experiments were conducted over three cycles, and results showed that lead ion adsorption declined from 100% to 46% in the second cycle and subsequently decreased to 36% in the third cycle [44].

The mechanism for DON adsorption by the microspheres was analyzed by SEM and FTIR analyses. Both physical adsorption and hydrogen bonding were involved. For physical adsorption, according to the SEM pictures for the SA/CMC-Ca microspheres, before DON adsorption, the internal surface of the SA/CMC-Ca microspheres was highly porous, which provided a large specific surface area for the adsorption of DON. This feature was similar to that observed for SA/CMC gel beads, which exhibited a larger specific surface area and porosity for the internal network structure after lead ion adsorption [30]. With the adsorption of DON, the SA/CMC-Ca microspheres inevitably adsorbed a small amount of the CSL components. These two factors made the pores smaller and color darker in the internal structure. Similarly, the adsorption of uranium and fluorine caused adsorbent surfaces to become rougher and more protruded [25]. Hydrogen bonding is one of the most significant types of adsorption. In a study, the formation of the hydrogen bond of the OH group of methanol with organic molecules containing COO- functional groups was modeled [45]. Both CMC and SA are rich in carboxyl groups [46,47]. From the FTIR analysis, the wavelength of O-H and COO- in the microspheres after the adsorption of DON was longer than that before the adsorption of DON. All these characteristics showed that the hydroxyl group contained in the molecule of DON was bonded to the carboxyl group in the microsphere adsorbent by hydrogen bonding.

In summary, we herein report for the first time, the adsorption of DON in CSL by the SA/CMC-Ca microsphere adsorbent.

## 4. Conclusions

To conclude, an SA/CMC-Ca microsphere adsorbent was successfully synthesized for DON adsorption in CSL via an emulsification approach. Microspheres can be regenerated and recycled for at least 155 times. These SA/CMC-Ca microsphere adsorbents can be potentially used in feeding industries to adsorb DON.

## 5. Materials and Methods

### 5.1. Chemicals

Sodium alginate, carboxymethyl cellulose sodium composite, and calcium chloride were purchased from Sinopharm Chemical Reagent (Shanghai, China). Soybean oil was supplied by Yihai Kerry Jinlongyu Cereals and Oils Food (Wuhan, China). Phosphate-buffered saline (PBS) was obtained from Solarbio Science & Technology (Beijing, China). Methanol and acetonitrile were supplied by Thermo Fisher Scientific (Shanghai, China). CSL was obtained from Cofco Biochemical (Jilin, China). Sodium hydroxide was purchased from Beijing Chemical Works (Beijing, China). DON (≥98% purity) was purchased from Sigma-Aldrich (St. Louis, MO, USA).

### 5.2. Preparation of Microspheres

The gelation buffer consisted of 20 g SA and 6 g CMC, mixed in 1 L deionized water (Ming Che Water Purification System; Merck Millipore, China). An emulsification process for preparing microspheres is shown in Figure 11. The solution was stirred with the emulsifier (Fluko FM200 Fluko Technology Development Co., Ltd., Raleigh, NC, USA) for 30 min until completely dissolved. After this, the solution was mixed and stirred with 2 L of soybean oil in a dispersing mixer (ESJ-500 ELE Mechanical & Electrical Equipment Co., Ltd., Shanghai, China) for 1 min at 11 rpm. Then, CaCl_2_ solution for 2L (2L deionized water + 40 g CaCl_2_) was added and stirred at 5 rpm for 1 h. After that, deionized water was used to clean microspheres three times and 1L of deionized water was used every time for cleaning. After each time, microspheres were sieved by cotton yarn. [48]. After the microspheres were collected, they were weighed up to 5 g each for downstream analyses.

### 5.3. DON Extraction and HPLC Detection

DON was extracted according to the instructions described in [49]. Briefly, Samples for DON-contaminated CSL were added to 4 mL deionized water, then vortexed for 10 min, and centrifuged (5840R Eppendorf Co., Ltd., Hamburg, Germany) at 4000 rpm for 5 min. The supernatant was filtered through a sterile 0.22 μm pore filter (NYLON66 Syringe Filters; Tianjin Branch Billion Lung Experimental Equipment Co., Ltd., Tianjin, China). Precisely, 1 mL filtered supernatant was added to 9 mL 1% PBS for DON purification on an immunological affinity column (Huaan Magnech Bio-Tech Co., Ltd., Beijing, China). The column was cleaned twice in 10 mL deionized water. After cleaning, 1 mL methanol was added and the eluent collected. This eluent was evaporated under nitrogen at 60 °C for 40 min (OA-HEAT Heating System-American Organomation Co., Ltd., Berlin, MA, USA). The dried extract was dissolved in 200 μL acetonitrile in water (1:9 *v*/*v*). The solution was passed through a sterile 0.22 μm pore filter.

DON was detected according to a previously reported method [49]. Acetonitrile in water (1:9, *v*/*v*) mobile phase was prepared and used for high-performance liquid chromatography (HPLC) (Waters, Milford, MA, USA) on a C18 column (150 mm × 4.6 mm, 5 μm; Agilent Technologies, Santa Clara, CA, USA). A UV detector (Waters, Milford, MA, USA) was set at a wavelength of 220 nm, and the running time was determined as 20 min. Stock standard solutions of DON at concentrations of 1 μg/mL, 2 μg/mL,3 μg/mL,4 μg/mL, 5 μg/mL, 6 μg/mL, 7 μg/mL, 8 μg/mL, 9 μg/mL, and 10 μg/mL were prepared with acetonitrile. The retention time for DON was approximately 17 min, identified by comparing retention time and peak areas with calibration to 4 μg/mL DON standard. Other DON standard solutions were also detected by HPLC. The peak area was recorded. With the DON standard concentration as the abscissa and the measured peak area as the ordinate, the standard curve was made. As shown in Figure 12, the obtained standard curve conformed to the linear regression equation of y = 18068x + 749.87 and R² = 0.9982. The results showed that there was a good linear relationship between the DON concentration and chromatographic peak area. DON levels in the samples were calculated by the standard curve of the DON.

### 5.4. Microsphere DON Adsorption Test

Microspheres (5 g) were placed in 10 mL of DON-polluted CSL, which was named the treatment group. DON-polluted CSL (10 mL) was used as a control group. All groups were tested in three biological replicates. The treatment and control groups were reacted for 72 h. After this, DON was extracted and HPLC was used to detect DON levels by the method showed in 5.3.

### 5.5. The Effect of CSL Volumes

Different volumes of 3 mL, 4 mL, 5 mL, 6 mL, 7 mL, 8 mL, 9 mL and 10 mL of CSL for three biological replicates were used to react with 5 g microspheres and then filter the CSL by a cotton sieve. Three biological replicates of original CSL were made in control (CK1, CK2, and CK3). After this procedure, DON was extracted and DON level was assessed by the HPLC method in 5.3.

### 5.6. The Effect of Reaction Times

Different reaction times of 1 min, 3 min, 5 min, 7 min, and 9 min were used to assess reaction times for 5 mL CSL polluted by DON with 5 g microspheres. All samples were tested in three biological replicates, and then CSL was filtered by a cotton sieve. Original CSL was made as a control. After this procedure, DON was extracted and the adsorption rates were assessed by the HPLC method in 5.3. The speed of DON adsorption is as follows:$$\mathrm{Speed}\;\mathrm{of}\;\mathrm{DON}\;\mathrm{adsorption}=\frac{c_{0}-c}{t}$$where c_0_ is the DON level in original CSL, c is the DON level in different reaction times of CSL, and t is the reaction time (minutes) of CSL.

### 5.7. The Effect of Time for Desorption

Desorption times of 1 min, 3 min, 5 min, 7 min, 10 min, and 60 min were tested by microspheres that had been regenerated 10 times. Each time, 5 mL DON-polluted CSL with 5 g microspheres was used. All samples were tested in three biological replicates, and the microspheres were filtered by a cotton sieve. The microspheres were soaked in 20 mL deionized water, and original CSL was used as a control. After this procedure, DON was extracted and the adsorption rates were assessed by the HPLC method in 5.3.

### 5.8. Detoxification of DON in Microsphere Cleaning Water

Microsphere cleaning water (40 mL) was collected and the pH was adjusted to 12 with NaOH. Two milliliters of cleaning water was taken in a 10 mL centrifuge tube and heated in a water bath (OLABO., Jinan, China) at 70 °C for 10 min, 30 min, and 60 min. All samples were tested in three biological replicates, DON was extracted, and the detoxification rate was assessed by the HPLC method in Section 5.3. The detoxification rate is as follows:$$\mathrm{detoxification}\;\mathrm{rate}=\frac{\left(c_{0}-c\right)}{c_{0}}\times 100\%$$where c_0_ is the DON level in original cleaning water and c is the DON level in cleaning water after treatment with NaOH.

### 5.9. Microspheres Recyclability Process

Microspheres (5 g) were reacted with 5 mL CSL polluted by DON for 5 min. After that, CSL was filtered by a cotton sieve and three biological replicates of original CSL (control) and treated CSL were prepared. After every fifth recycle time, DON was extracted and the DON level in CSL was assessed by the HPLC method in Section 5.3. After that, microspheres were cleaned in 20 mL deionized water for 5 min and the microspheres were filtered for a new recycle. All samples were recycled for 170 times.

### 5.10. SEM Analysis

Different states of microspheres (1 g) were prepared and cleaned twice in water. After cleaning, 3 mL ethanol in water (7:3 *v*/*v*) was added and the microspheres were fixed for 24 h. After this, samples were gradient-eluted in 70%, 80%, 90%, and 100% ethanol, each for 10 min. The samples were then dried on a Leica EM CPD030 automated critical point dryer (Leica Microsystems Inc, Wetzlar, Germany). Ultrathin sample sections were prepared using an ultramicrotome (Leica EM UC6) (Leica Microsystems Inc, Wetzlar, Germany). The surface morphology was observed using a HITACHI SU8010 scanning electron microscope (Hitachi High-Technologies Corporation, Tokyo, Japan).

### 5.11. FTIR Analysis

Samples for SA-Ca microspheres, SA/CMC-Ca microspheres, SA/CMC-Ca microspheres that adsorbed DON in CSL, and SA/CMC-Ca microspheres that desorbed DON in cleaning water were recorded on a BRUKER TENSOR 27 infrared spectrometer (Bruker Co., Ltd., Bremen, Germany) within the wavelength range of 600–4000 cm^−1^. The weight of each of the samples was 0.1 g.

### 5.12. Statistical Analysis

Differences between groups were evaluated by analysis of variance with the general linear model procedure in a completely randomized single-factor design using the Statistical Program for Social Sciences (SPSS) version 20. *F* tests were performed at the 0.05 level of probability. When a significant *F* value was detected, significant differences among the means were assessed with Duncan’s multiple range test.

## Figures and Tables

**Figure 1 toxins-12-00208-f001:**
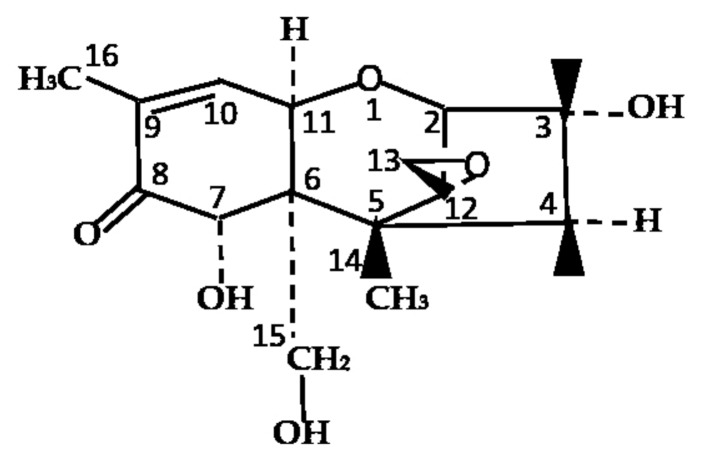
Chemical structure of deoxynivalenol (DON). Chemical formula: C_15_H_20_O_6_, molecular weight: 296.319 g/mol.

**Figure 2 toxins-12-00208-f002:**
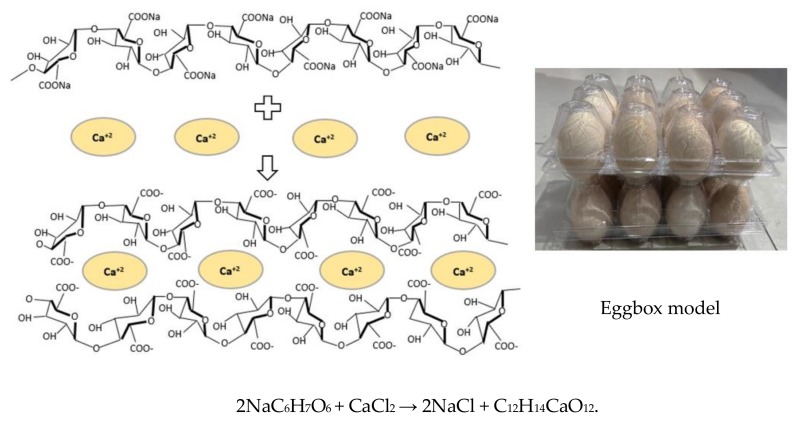
The formation mechanism for calcium alginate. Sodium alginate (NaC_6_H_7_O_6_) reacts with calcium chloride (CaCl_2_) to generate calcium alginate (C_12_H_14_CaO_12_), which is a gelatinous material. The two chemicals are rearranged, so they bond (like the eggbox model) to form a gelatinous (jello-like) substance.

**Figure 3 toxins-12-00208-f003:**
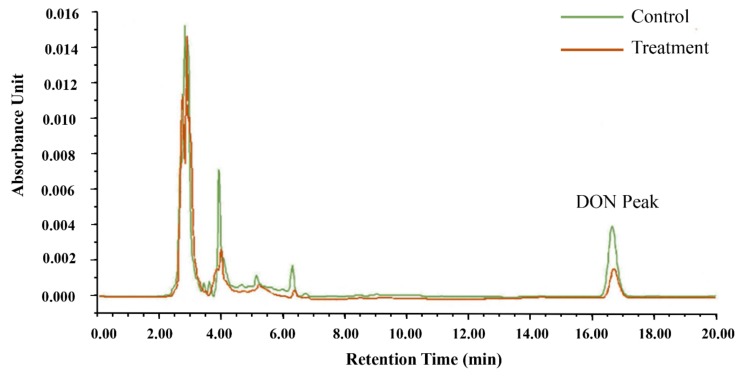
Treatment and control for the detection of DON in corn steep liquor (CSL) by HPLC.

**Figure 4 toxins-12-00208-f004:**
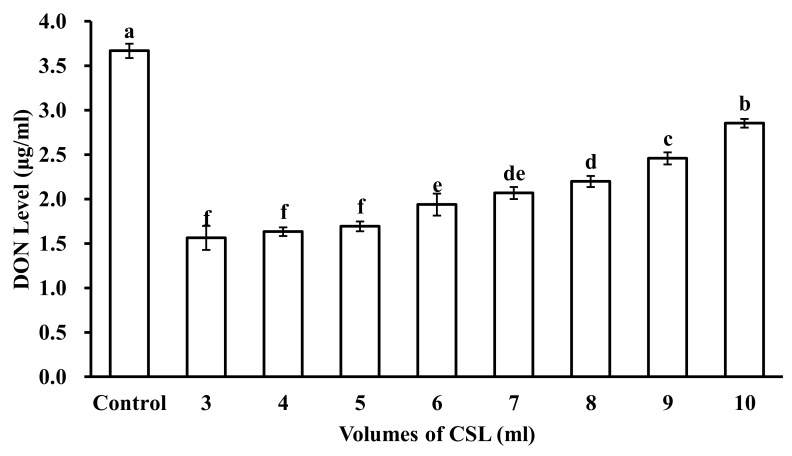
DON level in different volumes of CSL reacted with 5 mL microspheres. Control is DON level in original CSL. Values represent the means of three replicates and their standard errors. Different letters indicate significant differences between treatments (*p* < 0.05).

**Figure 5 toxins-12-00208-f005:**
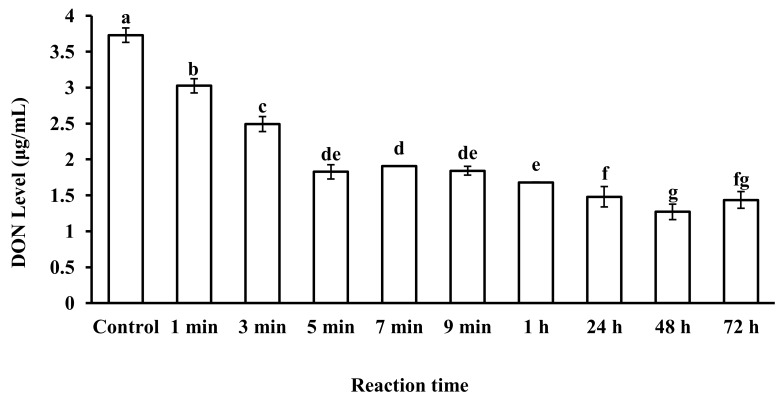
DON level in CSL after reaction with microspheres at different reaction times. Values represent the means of three replicates and their standard errors. Different letters for a, b, c, d, e, f, g indicate significant differences between treatments (*p* < 0.05).

**Figure 6 toxins-12-00208-f006:**
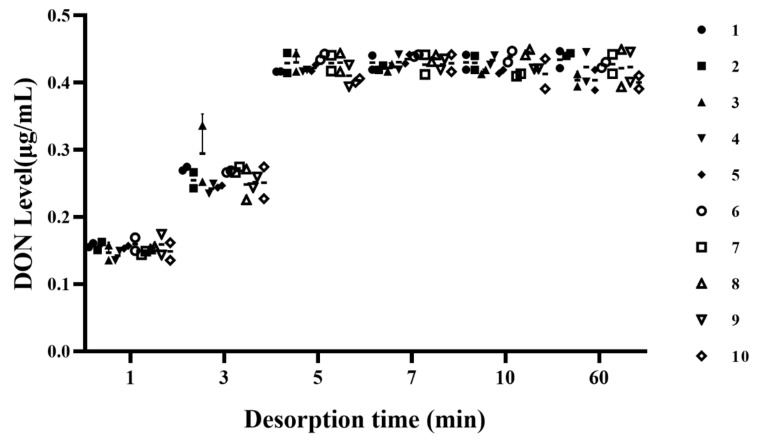
Effect of desorption time on DON level in deionized water for microspheres regenerated 10 times.

**Figure 7 toxins-12-00208-f007:**
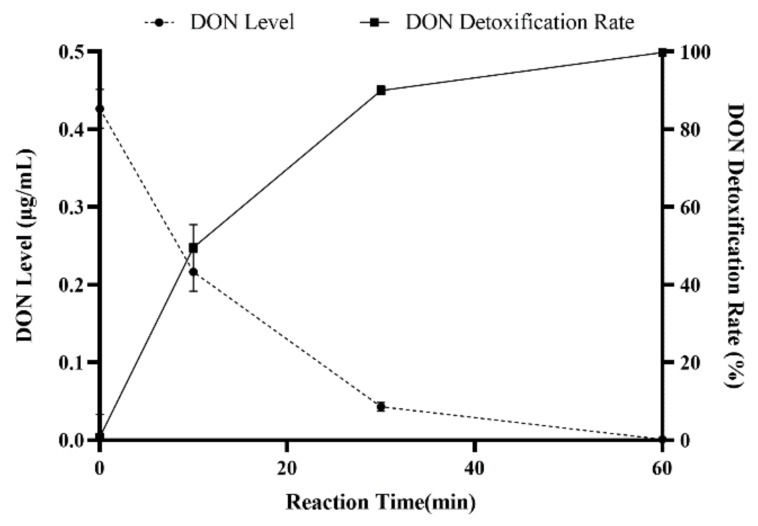
Detoxification of DON in contaminated water under alkaline high-temperature conditions. Values represent the means of three replicates and their standard errors.

**Figure 8 toxins-12-00208-f008:**
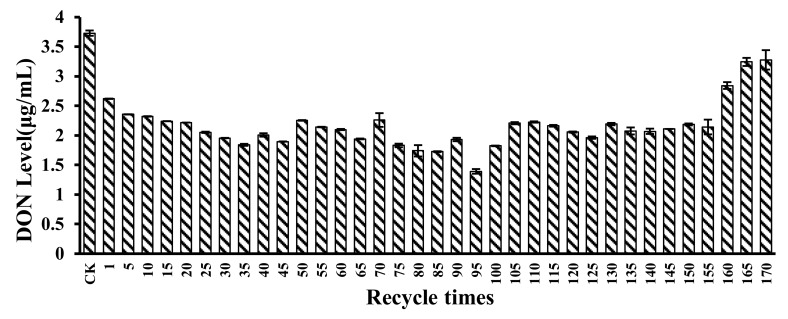
DON level in CSL after reaction with microspheres for 170 recycling times. The DON level in CSL was 3.73 μg/mL for the control. Values represent the means of three replicates and their standard errors.

**Figure 9 toxins-12-00208-f009:**
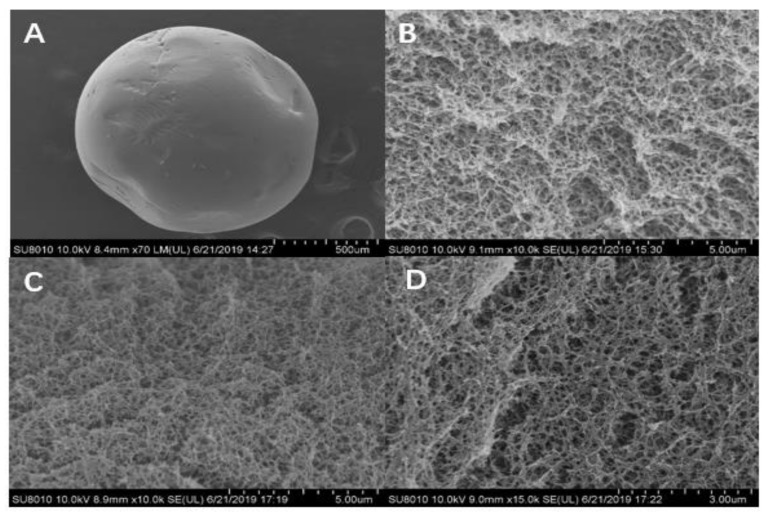
(**A**) The outer surface of an SA/CMC-Ca microsphere. (**B**) The internal structure of SA/CMC-Ca microspheres. (**C**) SEM of SA/CMC-Ca microspheres after adsorption of DON in contaminated CSL. (**D**) SEM of SA/CMC-Ca microspheres after cleaning with deionized water.

**Figure 10 toxins-12-00208-f010:**
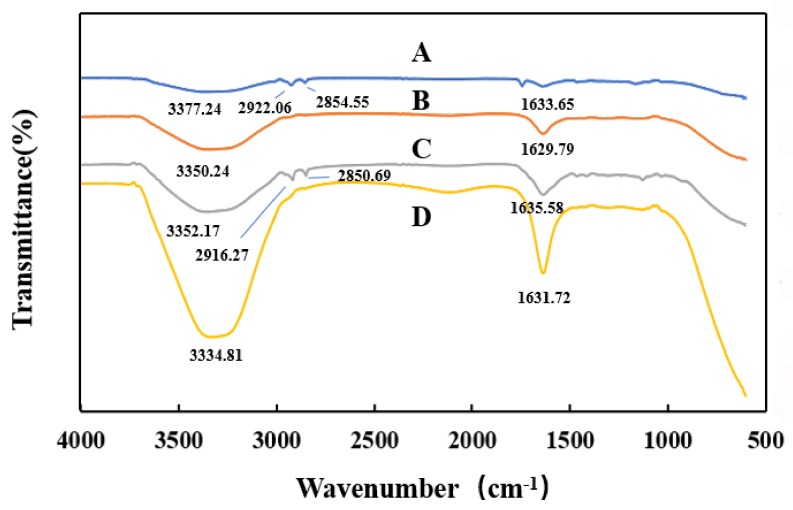
FTIR spectra of (**A**) SA-Ca microspheres, (**B**) SA/CMC-Ca microspheres, (**C**) SA/CMC-Ca microspheres that adsorbed DON, and (**D**) SA/CMC-Ca microspheres that desorbed DON.

**Figure 11 toxins-12-00208-f011:**
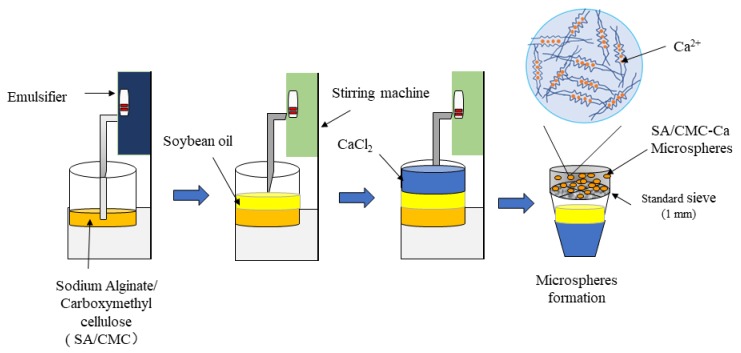
Schematic diagram of the microsphere preparation process.

**Figure 12 toxins-12-00208-f012:**
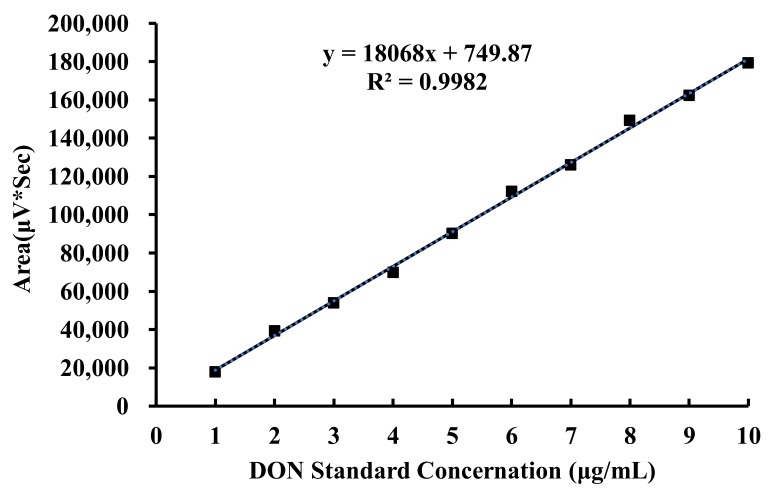
Standard curve of DON.

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
