# Peer review of "Adsorption of Deoxynivalenol (DON) from Corn Steep Liquor (CSL) by the Microsphere Adsorbent SA/CMC Loaded with Calcium"

_toxins, 2020, doi:10.3390/toxins12040208_

Round 1

Reviewer 1 Report

The changes and modifications in the revised manuscript are adequately done. 

Author Response

Dear Reviewer:

Thank you very much for your comments concerning our manuscript entitled “Adsorption of deoxynivalenol (DON) from corn steep liquor (CSL) by the microsphere adsorbent SA/CMC loaded with calcium” (ID: 738461).

According to another reviewer’s comments, we still have some language problems and some details about the manuscript not very clear. Hence, we invite the International Science Editing to revised some language, and we also ask some suggestions from doctor students in our lab.

We have finished this manuscript. If you have any questions regarding the revision of our manuscript,do not hesitate to contact us!

 A PDF attachment is added for the response to our revised paper! Please have a check.

Reviewer 2 Report

In this paper, the authors describe a dual-functional microsphere adsorbent comprising of an alginate/carboxymethyl cellulose sodium composite, loaded with calcium (SA/CMC-Ca) as an absorbent of DON from polluted corn steep liquor (CSL). Experiments were conducted on CSL volumes, reaction times, desorption times, and microspheres recyclability. The results were quite satisfactory and the authors propose this methodology as a way to address DON-contaminated CSL issues in animal feeds. However, this manuscript could be considered for publication in Toxins after the following MAJOR revisions:

Even though this research paper presents rather interesting results, the whole text needs extensive English editing because in most parts I was not able to understand what the authors were describing. In addition, some Figures numbers must be changed because they are repeated two and even three times. Finally, please add more information about the DON solution (purity, solvent).

Author Response

Dear Reviewer:

Thank you for your comments concerning our manuscript entitled “Adsorption of deoxynivalenol (DON) from corn steep liquor (CSL) by the microsphere adsorbent SA/CMC loaded with calcium” (ID: 738461). Those comments are all valuable and very helpful for revising and improving our paper, as well as the important guiding significance to our researches.

As your comment, the main problem for my manuscript is the language problem, so we invite the International Science Editing to revised some language and we also ask some suggestions from doctor students in our lab. We have finished this manuscript if you have any questions regarding the revision of our manuscript,do not hesitate to contact us!

A PDF attachment is added for the response to our revised manuscript! please have a check.

Round 2

Reviewer 2 Report

As I can see, the authors have done thorough English editing and corrected the grammatical mistakes in the revised manuscript. Thus the manuscript can be accepted in the present form.

This manuscript is a resubmission of an earlier submission. The following is a list of the peer review reports and author responses from that submission.

Round 1

Reviewer 1 Report

The authors describe the potential application of microspheres made up of alginate and carboxymethyl cellulose loaded with calcium to remove DON from corn steep liquor. In addition they describe how the effluents containing DON can be potentially decontaminated by the application of a bacterial DON degrading strain.

Apart from the introduction, language difficulties often prevent a clear understanding of the text - Are the authors claiming aBsorption or aDsorption, DON removal, DON degradation or DON detoxification? Even though the materials section contains lots of text, details are missing on the bacterial strain, HPLC methodology HPLC standards. etc. As standard solutions of known concentration were purchased from Sigma Aldrich, it should have been easy to quantify DON, instead of giving only % values. What are the limits of detection and limits of quantification when DON is extracted from a matrix like CSL? And are the losses due to the extraction protocol rather than the presence of the micro spheres in the CSL, unfortunately this control is missing, or has not been reported. The bacterial strain Devosia is not described or referenced at all. Not all Devosia do degrade DON, thus the species and the actual behavior towards DON should be tested and shown. Only if the metabolite is defined, detected, and known to be less toxic, a detoxification can be claimed. The micro spheres are poorly described, as one wonders if the pores created are large enough to contain a bulky substrate like DON, or how the interaction is believed to take place. The scale in the SEM images B, C, D differs, even though they are compared to each other. Lot´s of questions are not addressed at all, such as what else binds to the microspheres? How well do they distribute in the CSL? The application is at a ratio of 1: 1 weight per volume (5 g in 5 ml) is this really feasible at a large scale? Most references are used to underline assumed characteristics of the microsphere, but lack the according data. This manuscript, even though it is comprehensive, lacks relevant information for publication.

Reviewer 2 Report

Review Report

A brief summary

The study was aimed at to investigate the efficiency of the microsphere adsorbent SA/CMC loaded with calcium to adsorb deoxynivalenol (DON) from corn steep liquor (CSL) polluted by 3.60 μg/mL DON.

Broad comments

The experiments are logic and seemed to be carefully done. However, the paper is not thoroughly written. The introduction is very long with 45 cited references. In addition, there are mentioned some unnecessary information. The results were completely and systematically presented. There are stated some information in the results that belongs to materials and methods (e.g. pages 3 and 4 lines 110–114). In some other sections in results, the results are discussed although the manuscript has a discussion part (e.g. page 4 lines 129–131 and page 6 lines 168–169). The material and methods part is missing description of statistical methods. Therefore, the validity of the presented results or data cannot be correctly reviewed.